# Development and validation of small animal models for onchocerciasis and loiasis microfilaricide discovery

**Rene Bilingwe Ayiseh**[1], **Glory Enjong Mbah**[1¤a], **Elvis Monya**[1], **Emmanuel Menang Ndi**[1¤b], **Judy Sakanari**[2], **Sara Lustigman**[3], **Fidelis Cho-Ngwa**[1¤c]*

**1** ANDI Centre of Excellence for Onchocerciasis Drug Research, Biotechnology Unit, Faculty of Science, University of Buea, Buea, Cameroon, **2** Department of Pharmaceutical Chemistry, University of California, San Francisco, California, United States of America, **3** Lindsley F. Kimball Research Institute, New York Blood Center, New York City, New York, United States of America

¤a Current address: Department of Biology, Higher Teacher Training College (HTTC), University of Bamenda, Bamenda, Cameroon
¤b Current address: Institute of Molecular Biotechnology, University of Bodenkultur, Vienna, Austria
¤c Current address: National Higher Polytechnic Institute (NAHPI), University of Bamenda, Bamenda, Cameroon
* fidelis.cho@ubuea.cm

**Data Availability Statement:** All relevant data are within the manuscript and its Supporting Information files.

## Abstract

### Background

Onchocerciasis (river blindness) caused by the filarial worm *Onchocerca volvulus* is a neglected tropical disease that affects the skin and eyes of humans. Mass drug administration with ivermectin (IVM) to control the disease often suffers from severe adverse events in individuals co-injected with high loads of *Loa loa* microfilariae (mf). Thus loiasis animal models for counter-screening of compounds effective against onchocerciasis are needed, as are the corresponding onchocerciasis screening models. The repertoire of such models is highly limiting. Therefore, this study was aimed at developing and validating mf immunocompetent small animal models to increase tools for onchocerciasis drug discovery.

### Methodology/Principal findings

*O. ochengi* mf from cattle skin and *L. loa* mf from human blood were used to infect BALB/c mice and Mongolian gerbils, and IVM was used for model validation. *O. ochengi* mf were given subcutaneously to both rodents while *L. loa* mf were administered intravenously to mice and intraperitoneally to gerbils. IVM was given orally. In an 8-day model of *O. ochengi* mf in BALB/c mice, treatment with IVM depleted all mf in the mice, unlike the controls. Also, in a 2.5-day model of *L. loa* mf in BALB/c, IVM significantly reduced mf in treated mice compared to the untreated. Furthermore, the gerbils were very susceptible to *O. ochengi* mf and IVM eradicated all mf in the treated animals. In the peritoneal *L. loa* mf gerbil model, IVM reduced mf motility in treated animals compared to the controls. In a 30-day gerbil co-injection model, IVM treatment cleared all *O. ochengi* mf and reduced motility of *L. loa* mf. Both mf survived for up to 50 days in a gerbil co-injection model.

**Funding:** This study was supported by grants from the Bill and Melinda Gates Foundation through the Grand Challenges Exploration Initiative under grant ID OPP1098475 to FCN and OPP1017584 to FCN, JS, and SL. The funders had no role in study design, data collection and analysis, decision to publish, or preparation of the manuscript.

**Competing interests:** The authors have declared that no competing interests exist.

## Conclusions/Significance

We have developed two immunocompetent small animal models for onchocerciasis and loiasis that can be used for microfilaricide discovery and to counter-screen onchocerciasis macrofilarides

### Author summary

Onchocerciasis (river blindness) caused by the filarial worm *Onchocerca volvulus* is a severe skin disease and affects the eyes leading to vision loss. Drugs that are effective against *O. volvulus* should be ineffective against another filarial worm *Loa loa*, as severe adverse events has been reported in treated co-infected individuals. Thus, animal models are needed to screen for compounds effective against onchocerciasis but not against the *Loa loa* microfilariae (mf). *O. ochengi* mf from cattle skin and *L. loa* mf from human blood were used to infect BALB/c mice and Mongolian gerbils, and ivermectin (IVM) was used for model validation. *O. ochengi* mf survived both animals and treatment with IVM significantly reduced the number of mf. In a BALB/c *L. loa* mf model, IVM significantly reduced mf in treated mice compared to the untreated. *L. loa* mf survived in peritoneum of gerbils. A gerbil co-injection model (with both *O. ochengi* and *L. loa* mf) was developed and both mf survived for up to 50 days in this model. In conclusion, we have developed two immunocompetent small animal models for onchocerciasis and loiasis that can be used for microfilaricide discovery.

## Introduction

Onchocerciasis also known as river blindness is a neglected tropical disease that affects the skin and eyes. It is caused by the filarial worm *Onchocerca volvulus* and transmitted by the black fly of genus *Simulium*. According to WHO, the global burden of onchocerciasis in 2017 was 20.9 million people infected with 14.6 million presenting skin disease and 1.15 million having vision loss [1]. More than 99% of those afflicted live in sub-Saharan Africa. At the moment, the strategy to eliminate the disease is mass drug administration with the drug of choice, ivermectin (IVM) which can break the transmission cycle. However, this effort has suffered many setbacks such as the development of severe adverse events in people co-injected with high loads of *Loa loa* microfilariae (mf) [2], and the possible emergence of IVM drug resistance [3,4].

The disease loiasis, otherwise considered a benign infection, has attracted much attention mainly in the context of hindering mass drug administration of IVM in treating onchocerciasis. It is estimated that at least 10 million people are infected with *L. loa* with 14.4 million people living in high risk regions [5,6]. Loiasis is characterized by Calabar swellings (oedema in the limbs), pruritus and subconjunctival migration of adult filariae [7]. Some studies have shown atypical clinical manifestations such as cardiac, respiratory, renal and neurological pathologies [8–10], and an association between high *L. loa* mf load and increased human mortality [11,12].

The call to end the epidemic of neglected tropical disease by 2030, by the WHO may be challenging for onchocerciasis without more understanding of the parasite. Animal models play a vital role in understanding parasite biology and can also be used for drug development. Any compound that is effective against onchocerciasis should not be active against *L. loa* mf to

avert severe adverse events. Thus, loiasis animal models are needed to counter-screen compounds effective against onchocerciasis.

Apart from humans, only non-human primates are susceptible to the full life cycle of *O. volvulus* [13]. Immunocompetent rodents are non-permissive to the *O. volvulus*, thus the severe combined immuno-deficient (SCID) and humanized mice have been used to enhance susceptibility [14,15]. *O. ochengi* which is the closest relative to *O. volvulus* [16,17] has been exploited for about 30 years in its natural bovine system for drug and vaccine development [18]. Although such a model is very expensive to run given its size, skin of infected cattle that have been slaughtered serves as a source for adult worms and mf. These have been used for many *in vitro* studies and to develop smaller animal models for onchocerciasis. The gerbil was shown to harbour adult male *O. ochengi* [19] and this model has been validated, and is used for drug testing [20,21]. Onchocercomata implanted in severe combined immuno-deficient (SCID) mice show motile male and female worms 7 days post-implantation [19]. The Syrian hamster has been developed as a model for *O. ochengi* mf [22].

The life cycle of *L. loa* can be maintained in non-human primates–the drill, baboon and patas monkey [23,24]. Although adult *L. loa* transplanted into gerbils produce mf [25], it has not been successfully established as a small animal model [26,27]. The infective stage (L3) of *L. loa* developed to young adults in mice with impaired Th2 cytokine signaling [28] while it reached adulthood and produced mf in lymphopenic mice [29]. Interestingly, *L. loa* mf could be detected in cardiac blood of immunocompetent BALB/c mice 7 days post-infection [29].

Most of the models for onchocerciasis and loiasis described above are either non-human primates which limit the throughput for compound screens, or rodents with impaired immune system. Some drugs act through the immune system, so it is preferable not to use immuno-compromised animals in preclinical studies. Therefore, this study was aimed at developing and validating mf immunocompetent small animal models to increase tools for onchocerciasis drug discovery and research. Immunocompetent small animal models for either loiasis or onchocerciasis or both diseases were developed and validated.

## Methods

### Ethics statement

Ethical approval N˚ 964–09 for the use of human blood samples with *L. loa* mf in this study was obtained from the Institutional Review Board of the Faculty of Health Science of the University of Buea, while administrative clearance N˚ 631–06.14 was obtained from the Cameroon Ministry of Public Health in Yaoundé. *L. loa* endemic areas of Edea and Mfou health districts of the Littoral and Centre regions of Cameroon, respectively, were targeted. Participants included those who consented to participate in the study as *L. loa* parasite donors after proper understanding of the intended study. Individuals eligible for participation were adults of both sexes, >21 years of age, in good health without any clinical condition requiring long term treatment. Participants signed an informed consent form before taking part in the study. Exclusion criteria included an mf load of <5,000/ml of blood and use of anti-filarial therapy. A maximum of 20 ml of venous blood was collected from eligible individuals at most 24 hours before *L. loa* mf isolation.

The animal protocol (UB-IACUC No 010/2019) was approved by the University of Buea Animal Care and Use Committee, following recommendations from the 'Guide for the Care and Use of Laboratory Animals', 8th edition by the National Research Council, USA. This *in vivo* study was reported in accordance with the ARRIVE Guidelines for reporting animal research.

## Experimental animals

Mongolian gerbils (*Meriones unguiculatus*) were purchased from Charles River (France). These animals together with home-bred BALB/c mice were maintained in a conventional animal house at the Biotechnology Unit of the University of Buea. Similar numbers of both males and female animals were used. Animals were given food and water *ad libidum*. Animals were aged 8–15 weeks at the time of use in the experiments.

## Extraction of *O. ochengi* and *L. loa* mf

**O. ochengi mf isolation.** This was done following a method we described previously [30], with slight modifications. Briefly, fresh pieces of nodule-containing umbilical cattle skin were purchased from a local slaughter house, washed repeatedly with tap water until all dirt was removed, and rinsed with distilled water. The skin was towel-dried and sterilized with 70% ethanol, allowed to dry in a laminar flow hood and firmly attached to an autoclaved cylindrical-shaped wooden block. Using sharp razor blades, criss-cross cuts were made into the skin and submerged in ICM (incomplete culture medium: RPMI-1640 containing penicillin, streptomycin and amphotericin B) for 4 hours, after which, the medium was centrifuged at 700xg for 20 minutes using an Eppendorf 5810R centrifuge (Eppendorf, Germany) to concentrate the mf. After viability check, the number of mf were counted and adjusted to the required number (4,000–22,725 in 200–1,000 μl of ICM) for experimental infection.

**L. loa mf isolation.** one gram of medium-grain Sephadex G-50 (Pharmacia Biotech, Uppsala, Sweden) was weighed and allowed to swell overnight in 20 ml of distilled water at 4°C. Using a 25 ml serological pipette as column, the swollen gel (after attaining room temperature) was transferred and allowed to settle to a bed volume of 11 ml. This column was equilibrated with 2 bed volumes of ICM, which was also the elution buffer. The number of *L. loa* mf per ml of blood was determined microscopically. Then, 1 ml of the infected blood was loaded on the column and allowed to completely enter the gel bed before elution began. The red coloration due to red blood cells was allowed to flow through the gel completely before fraction collection started. A total of 30 ml of eluate was collected for every ml of blood loaded. Fractions were centrifuged at 700xg for 20 minutes using the Eppendorf 5810R centrifuge, supernatants discarded, pellets re-suspended in ICM, pooled and centrifuged as before. Sediments obtained after the second centrifugation were re-suspended in ICM and washed on 20% percoll at 100xg for 5 minutes, thrice using the same centrifuge. After the third wash, the procedure was repeated twice with 1x PBS in place of percoll. Sediments were finally re-suspended in a small volume of ICM, and *L. loa* mf counted using a microscope.

## Experimental infection with *O. ochengi* mf

Isolated mf were administered subcutaneously (sc) at the nape of gerbils and mice using a 27G needle syringe. The number of mf per animal depended on the number of mf obtained after isolation and ranged from 4,000 to 22,725 per animal, in 200 μl to 1 ml of ICM. Mice and gerbils were sacrificed 8 and 30 days post-infection, respectively, for mf analyses.

## Experimental infection with *L. loa* mf

BALB/c mice were briefly warmed under 40-60W bulb for 5 minutes to make the tail veins more visible. Mice were intravenously infected with 12,000 to 40,000 mf (depending on number of mf obtained after isolation) in 200 μl ICM. Gerbils were intraperitoneally infected with 16,000 isolated *L. loa* mf in 1 ml of ICM since the tail veins are not visible to allow for

intravenous injection. Mice and gerbils were sacrificed 2.5 and 30 days post-infection, respectively, for mf analyses.

## Co-injection of gerbils with mf of *L. loa* and *O. ochengi*

In co-injection experiments, gerbils received 1 ml of whole human blood containing 5,500–30,000 *L. loa* mf intraperitoneally. After one day, each animal received 5,000 or 4,000 isolated *O. ochengi* mf in 200 µl of ICM, subcutaneously at the nape. Animals were sacrificed 25, 40 or 50 days post *L. loa* infection and analyzed for mf recovery and motility.

## Drug administration

Animals were treated with ivermectin (Sigma Aldrich, Germany) by oral gavage. To validate *O. ochengi* mf models, animals were given 150 µg/kg body weight IVM (which is the dose give to humans) 1 and 3 days post-infection for mice and gerbils respectively. To validate the *L. loa* mf models, mice received 1 or 10 mg/kg body weight IVM 12 hours post mf infection, while gerbils got 150 µg/kg body weight IVM 7 days post-infection. For the co-injection gerbil model, two doses of 0.5 mg/kg IVM was administered 8 hours apart 7 days post *L. loa* mf infection. All control gerbils received the vehicle (1% DMSO in PBS).

## Checking animals for presence of mf

*O. ochengi* **mf.** Animals were sacrificed by by cervical dislocation, the fur on the ears and approximately 0.5 cm around the ears were carefully shaved to recover *O. ochengi* mf. Each ear was removed, rinsed in distilled water, towel-dried and sterilized with 70% ethanol. Each pair of ears was carefully minced into very small pieces and incubated in 5 ml of ICM at 37˚ C for 4 hours under sterile conditions. During preliminary tests, the entire animal skin was shaved and the skin of other body parts, including the fore limbs, hind limbs, head, tail and trunk were minced into tiny pieces and also examined for mf. After incubation, the total number of mf was obtained by spreading five 100 µl portions of ICM from each well on petri dishes and observing microscopically to count the mf.

*L. loa* **mf.** animals were sedated by intraperitoneal administration of 80/5 mg/kg body weight ketamine/xylazine before cardiac blood collection. *L. loa* mf was analysed in cardiac blood of mice and gerbils by staining 2x50 µl blood with 10% Giemsa. For Giemsa staining, 50 µl of blood was spread on a slide and allowed to air-dry for 10 minutes. The dried blood was covered with 10% Giemsa for 20 minutes and rinsed with running tap water. Slides were allowed to dry and *L. loa* mf were counted under the microscope. To determine the total number of mf in the blood, animals were weighed before sacrifice and the formula (blood volume (ml) = 0.06 X body weight (g) + 0.77) was used to estimate total blood volume [31]. Using the total blood volume, the total number of *L. loa* mf in the blood of each animal could be extrapolated from the number obtained under the microscope. Mice heart and lungs were removed and chopped into tiny pieces and each set incubated in 3 ml of culture medium in 12-well plates at 37˚C for 30 minutes. After incubation, 5x100 µl portions of the medium used to incubate the chopped organs in culture were observed microscopically for the presence of *L. loa* mf.

After cardiac blood was collected, each gerbil was sacrificed by cervical dislocation and the peritoneum was washed with appropriate volumes (20–30 ml) of PBS, depending on the size of the animal. The number of *L. loa* mf in 500 µl was counted and each scored for motility using an inverted microscope. Motility scores of 0% (complete inhibition of motility), 25% (only head or tail moving or vibrating), 50% (worm sluggish), 75% (little change in motility), or 100% (no observable inhibition of motility) were assigned to mf recovered from each

animal. The proportion of mf at each motility score was obtained by dividing the number of mf at that motility score by the total number of mf and multiplying by 100.

## Data analyses

Data were analysed using GraphPad Prism 5.0 software and statistical analyses were performed using Mann-Whitney non-parametric test for 2 groups of data or one-way ANOVA and Bonferroni's multiple comparison post-test for 3 or more groups of data. Data presented show mean±SEM. Data were considered to show statistical significant difference when $p<0.05$.

## Results

### *O. ochengi* microfilaria models

In an exploratory experiment (Table 1), three gerbils examined had mf in all the different parts of the skin (forelimbs, hind limbs, earlobes, tail, head and trunk). However, most of the mf (49.05±11.58%) were found in the earlobes, making it a predilection site. Thus, subsequent experiments focused on analyzing mf just in the earlobes. Following the administration of 22,725 *O. ochengi* mf, the number in the earlobes decreased from 1,173±135 on day 14 to 188 ±65 on day 30. No pathology was observed in the tissues or the whole body of the infected gerbils. For the validation experiments, treatment of gerbils with IVM completely eliminated *O. ochengi* mf (Fig 1A), whereas the control animals had 164±79 mf in the earlobes, which is 2.05% of the 8,000 mf injected.

Previously, we showed that BALB/c mice could not retain *O. ochengi* mf for 30 days [22]. Therefore, mice were sacrificed 8 days post-infection to improve on mf recovery. As with the gerbils, IVM eliminated *O. ochengi* mf in BALB/c mice (Fig 1B) whereas control animals had 42±14 mf, representing 0.83% of the 5,000 mf injected.

### *L. loa* microfilarial models

Assessment of *L. loa* mf in BALB/c mice 8 days post-infection showed mf in cardiac blood, heart and lung tissues (Table 2). On average, just 0.81% of the injected mf could be recovered from these 3 samples and most of the mf were found in the lungs. No pathology was seen in the tissues or the whole body of the infected mice.

Since most of the mf were found in the lungs of mice, the survival of *L. loa* mf therein was assessed from 2.5, 5, 6 and 8 days post-infection. After 2.5 days, 14.00±1.57% of injected mf were found in the lungs which significantly decreased to 1.85±0.61, 1.92±0.67 and 0.43±0.22% on days 5, 6 and 8 post-infection respectively (Fig 2A). The percentage of mf recovered in the

**Table 1. Susceptibility of gerbils to *O. ochengi* mf.** Each gerbil was subcutaneously infected at the nape with 22,725 *O. ochengi* mf. Gerbils were analyzed 14 or 30 days post-infection for *O. ochengi* mf in the ear lobes alone, or including the skin of other parts.

| Gerbil | Sex (M/F) | Duration between mf injection and sacrifice (days) | N° of mf in fore limbs | N° of mf in hind limbs | N° of mf in ear- lobes | N° of mf in tail | N° of mf in head | N° of mf in the trunk | Total mf recovered (% recovered) | % mf recovered in ear lobes relative to total recovery |
|---|---|---|---|---|---|---|---|---|---|---|
| 1 | M | 14 | 100 | 67 | 1000 | 83 | 233 | 50 | 1533 (6.75) | 65.19 |
| 2 | F | 14 | nd | nd | 1440 | nd | nd | nd | nd | nd |
| 3 | F | 14 | nd | nd | 1080 | nd | nd | nd | nd | nd |
| 4 | M | 30 | 7 | 13 | 315 | 53 | 80 | 100 | 569 (2.50) | 55.36 |
| 5 | M | 30 | 27 | 160 | 150 | 73 | 13 | 140 | 564 (2.48) | 26.60 |
| 6 | F | 30 | nd | nd | 100 | nd | nd | nd | nd | nd |

nd; not done

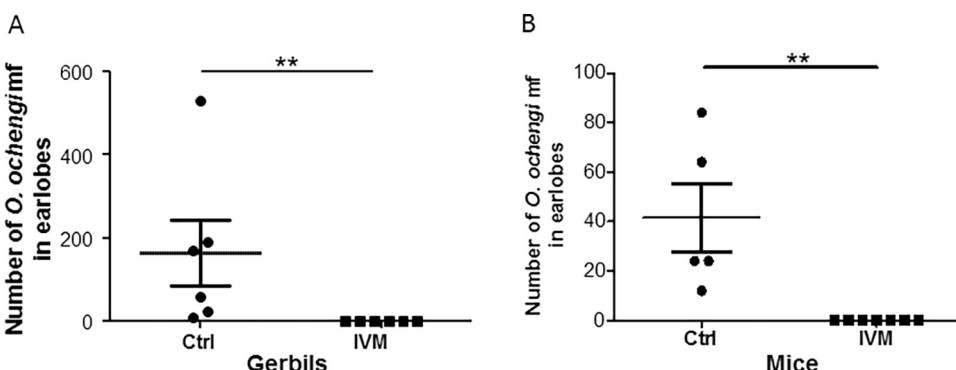

**Fig 1. Ivermectin (IVM) completely eliminates *O. ochengi* mf in gerbils and BALB/c mice.** (A) *O. ochengi* mf (8,000) were injected sc at the nape of each of 12 gerbils and 6 were treated by oral gavage on day 3 with 150 μg/kg body weight IVM. Control (Ctrl) animals received vehicle (1% DMSO). Animals were sacrificed 30 days post-infection and analyzed for mf in the earlobes, n = 6/group. **P<0.01. (B) *O. ochengi* mf (5,000) were injected sc at the nape of each of 12 mice. Seven were treated with 150 μg/kg body weight IVM by oral gavage 24 hours post-infection. Control animals received vehicle (1% DMSO in PBS). Animals were sacrificed 8 days post-infection and analyzed for mf in the earlobes. Male and female animals were used. n = 5-7/group, **P<0.01.

heart tissue (11.07±3.11%) was slightly lower mf than that in the lungs (14.00±1.57%) 2.5 days post-infection (Fig 2B). The variation in mf recovery was higher in the heart than in the lungs at 2.5 days post-infection.

Treatment of BALB/c mice with a single dose of IVM 12 hours after infection, significantly decreased the number of *L. loa* mf in the lungs and in cardiac blood after 2 days (Fig 3). The number of mf in the lungs was significantly reduced from 4,258±895 in the control group to 657±124 and 441±105 in the 1 and 10 mg/kg body weight IVM groups respectively (Fig 3A). This signifies 84.56 and 89.65% reduction of *L. loa* mf in the groups that received 1 and 10 mg/kg body weight IVM respectively. Similarly, mf in cardiac blood dropped significantly from 85 ±25 in the control mice to 16±4 and 5±1 in the 1 and 10 mg/kg body weight IVM groups respectively (Fig 3B). This represents 81.63 and 93.76% drop in mf in the 1 and 10 mg/kg body weight IVM groups respectively. In this experiment, the number of mf in the lungs and cardiac blood of control animals was 10.64±2.24 and 4.24±1.32% of the total mf injected.

Several exploratory studies with the gerbils following intraperitoneal infection with *L. loa* mf rarely showed mf in cardiac blood. Large numbers of *L. loa* mf were found in the

**Table 2. *L. loa* microfilariae are recovered in the blood, lungs and heart of BALB/c mice.** Mice were intravenously infected with 12,000 *L. loa* mf and analyzed on day 8 post-infection.

| Mouse | Sex (M/F) | N° of mf in cardiac blood | N° of mf in lungs tissue | N° of mf in heart tissue | Total mf recovered (% recovered) |
|---|---|---|---|---|---|
| 1 | F | 6 | 8 | 4 | 18 (0.15) |
| 2 | F | 0 | 0 | 4 | 4 (0.03) |
| 3 | F | 14 | 60 | 20 | 94 (0.78) |
| 4 | F | 2 | 0 | 16 | 18 (0.15) |
| 5 | M | 47 | 100 | 8 | 155 (1.29) |
| 6 | M | 2 | 0 | 0 | 2 (0.02) |
| 7 | M | 2 | 0 | 8 | 10 (0.09) |
| 8 | M | 0 | 72 | 8 | 80 (0.67) |
| 9 | F | nd | 468 | 24 | 492 (4.10) |
| **Average** | | 9.13 | 78.67 | 10.22 | 97.00 (0.81) |

nd; not done

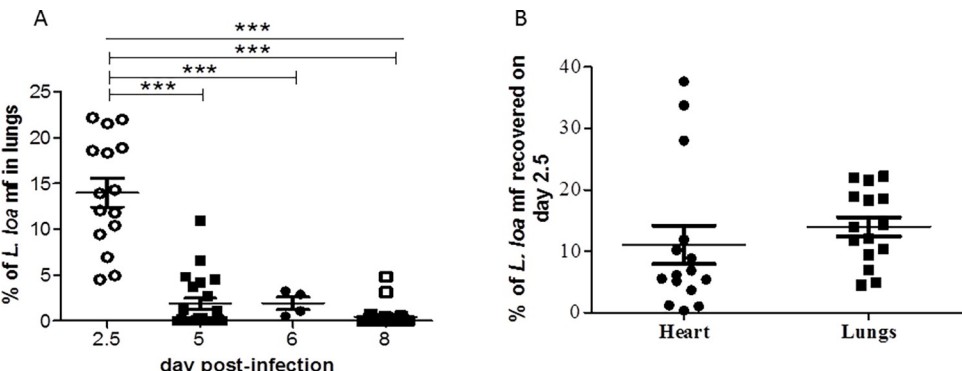

**Fig 2. Recovery of *L. loa* mf in BALB/c mice.** (A) Mice were infected intravenously with *L. loa* mf and the lungs were analyzed on various days post-infection. (B) *L. loa* mf analyzed in the heart and lung tissues of mice 2.5 days post-infection. Male and female mice were used. ***p<0.001.

peritoneum which became the targeted site for mf recovery. Treatment of *L. loa*-infected ger-bils with IVM (150 μg/kg body weight) led to a non-significant reduction from 5,478±1,388 in the control group to 3,665±1,653 (Fig 4A) mf in the treated group 30 days post-infection. This means there was 33.10% decrease in number of mf between the control and the treated group. The number of mf in the control group was 34.24% of the 16,000 mf administered. IVM decreased the motility of the worms as 68.58±9.01% of mf in the peritoneum of controls had 100% motility unlike 22.50±10.10% in the treated group (Fig 4B). Therefore, IVM treatment reduced the viability of the *L. loa* mf.

## *O. ochengi* and *L. loa* mf gerbil co-injection model

The gerbil was further assessed as a model for simultaneous infection with *O. ochengi* and *L. loa* mf, and treated with IVM (two doses of 0.5 mg/kg body weight). Similar to the previous results, treatment with IVM non-significantly decreased the number of *L. loa* mf from 2,474 ±506 in the control group to 1,152±605 which is a reduction of 55.44% (Fig 5A). IVM treat-ment equally had an impact on *L. loa* mf motility. In control animals, 75.25±17.67% of the mf were at 100% motility compared to 25.00±19.76% in the IVM-treated gerbils (Fig 5B).

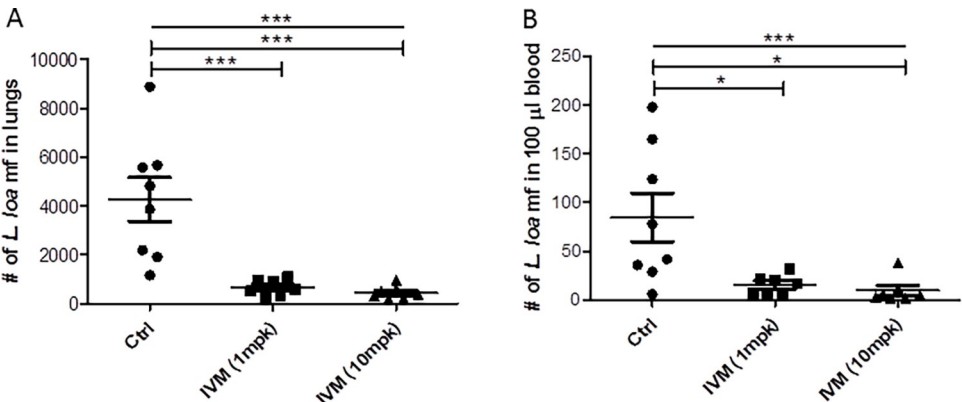

**Fig 3. Ivermectin significantly decreases *L. loa* mf in BALB/c mice.** Mice (males and females) were infected intravenously with 40,000 *L. loa* mf. After 12 hours, mice were orally treated with 1 or 10 mg/kg body weight IVM. Control animals received vehicle (1% DMSO in PBS). Mice were analyzed 2.5 days post-infection for *L. loa* mf in (A) Lungs, and (B) Cardiac blood. *P<0.05, ***P<0.001.

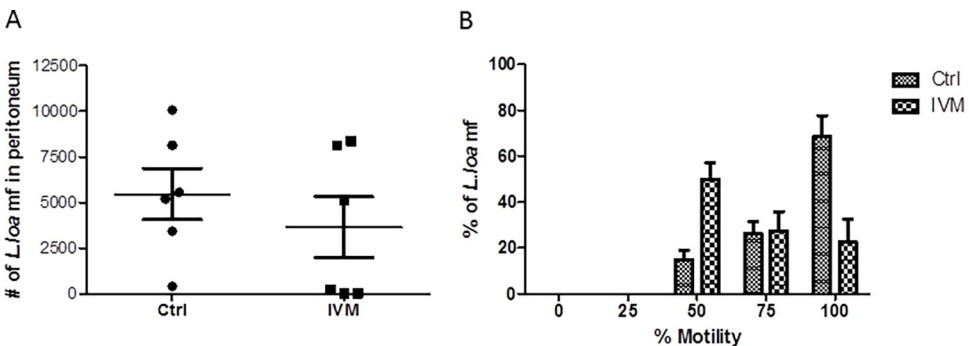

**Fig 4. Ivermectin reduces *L. loa* mf viability in gerbils.** Each of 12 animals (males and females) was infected intraperitoneally with 16,000 isolated *L. loa* mf and 6 of them were treated with 150 µg/kg body weight IVM by oral gavage on day 7. Control animals received vehicle (1% DMSO in PBS). Animals were sacrificed on 30 days post-infection and analyzed for mf. (A) Number of *L. loa* mf recovered from the peritoneum. (B) Proportion of *L. loa* mf from the animal peritoneum at different percent motility.

Treatment with IVM completely eliminated *O. ochengi* mf in the gerbils whereas control animals had 30±3 mf in the earlobes (Fig 5C). Thus, IVM treatment completely cleared *O. ochengi* mf, and reduced the number and motility of *L. loa* mf in co-injected gerbils.

Lastly, the gerbil co-injection model was assessed for both mf 40 and 50 days post-infection (Table 3). Forty days post-infection, 34.60±4.86% (10,379±1458 of 30,000 injected) *L. loa* were found in the peritoneum of gerbils and 7.09±3.49% (355±175 out of 5,000 injected) *O. ochengi* were present in the earlobes. After 50 days of infection, 33.69±10.77% (7,738±2154 of 20,000 injected) *L. loa* were found in the peritoneum and 3.88±0.25% (194±13 out of 5,000 injected)

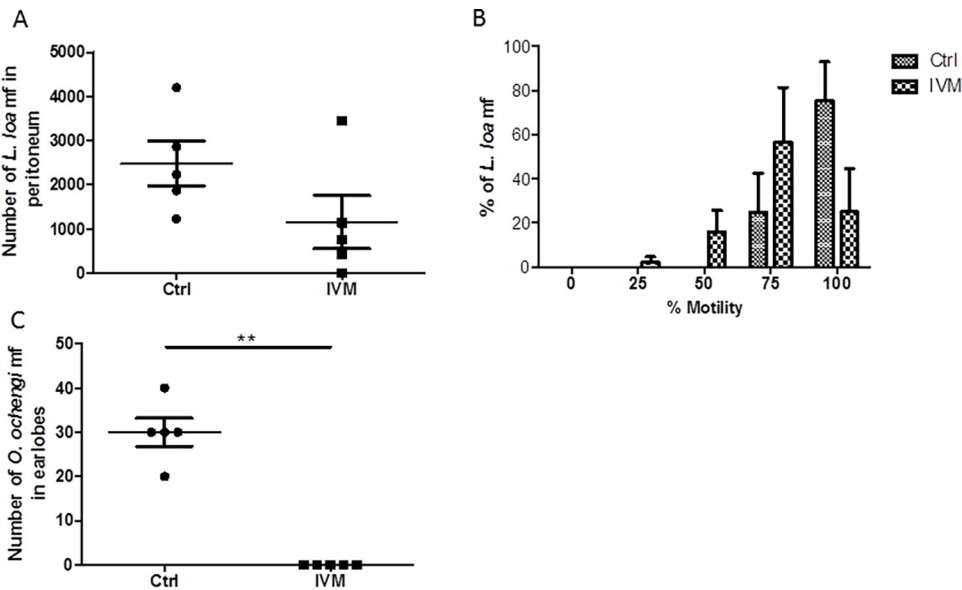

**Fig 5. Ivermectin reduces *L. loa* mf viability and eliminates *O. ochengi* mf in a gerbil co-injection model.** Each of 10 gerbils (males and females) was infected intraperitoneally with 5,500 *L. loa* mf and after 24 hours, they were subcutaneously infected with 4,000 *O. ochengi*. Five gerbils were treated on day 7 post *L. loa* infection with two doses of 0.5 mg/kg body weight IVM 8 hours apart by oral gavage. Control animals received vehicle (1% DMSO in PBS). Animals were sacrificed 25 days post *L. loa* infection and analyzed for mf. (A) Number of *L. loa* mf recovered from the peritoneum. (B) Proportion of *L. loa* mf from the animal peritoneum at different percent motility. (C) Number of *O. ochengi* mf in earlobes. **P<0.01.

**Table 3. Co-injection of gerbils with *L. loa* and *O. ochengi* mf for 40 and 50 days.** Male and female gerbils were intraperitoneally infected with the 30,000 or 20,000 *L. loa* mf and one day after, they were subcutaneously infected at the nape with 5,000 *O. ochengi* mf. Animals were analyzed 40 or 50 days post *L. loa* mf infection for both mf populations.

| Gerbil | Duration between mf injection and sacrifice (days) | *L. loa* | | | *O. ochengi* | |
|---|---|---|---|---|---|---|
| | | Number of mf injected | Number of mf in the peritoneum (%) | Number of mf in cardiac blood | Number of mf injected | Number of mf from earlobes (%) |
| 1 | 40 | 30,000 | 15,000 (50.00) | 0 | 5,000 | 114 (2.28) |
| 2 | 40 | 30,000 | 8,300 (27.67) | 0 | 5,000 | 342 (6.84) |
| 3 | 40 | 30,000 | 5,700 (19.00) | 0 | 5,000 | 414 (8.28) |
| 4 | 40 | 30,000 | 6,600 (22.00) | 0 | 5,000 | 42 (0.84) |
| 5 | 40 | 30,000 | 13,500 (45.00) | 0 | 5,000 | 34 (0.68) |
| 6 | 40 | 30,000 | 14,400 (48.00) | 0 | 5,000 | 186 (3.72) |
| 7 | 40 | 30,000 | 9,090 (30.30) | 60 | 5,000 | 1350 (27.00) |
| 8 | 50 | 20,000 | 10,950 (54.75) | 0 | 5,000 | 222 (4.44) |
| 9 | 50 | 20,000 | 1,750 (8.75) | 0 | 5,000 | 204 (4.08) |
| 10 | 50 | 20,000 | 9,650 (48.25) | 0 | 5,000 | 188 (3.76) |
| 11 | 50 | 20,000 | 4,600 (23.00) | 0 | 5,000 | 162 (3.24) |

*O. ochengi* were present in the earlobes. The *L. loa* mf at these time points were still viable, however most were at 50% motility (Fig 6). Only one out of the 11 animals analyzed had mf in cardiac blood. These results show that *O. ochengi* and *L. loa* mf can survive in gerbils at least 50 days post-infection.

## Discussion

There are a very limited number of animal models available to study onchocerciasis and loiasis. These models are mainly non-human primates or immuno-deficient rodents which are limiting for a robust exploration of the biology of the parasites that cause these diseases. While some drugs act directly on the filarial parasite, others exploit the immune system to be effective. This study was therefore aimed at developing immunocompetent small laboratory animal models to study these parasites with a prime focus on the microfilariae. We were able to successfully develop two small animal models using BALB/c and Mongolian gerbils, and validated the effect of IVM on microfilariae survival for both onchocerciasis and loiasis. In addition, the gerbil model was further advanced for testing survival of co-injected *O. ochengi* and *L. loa* mf.

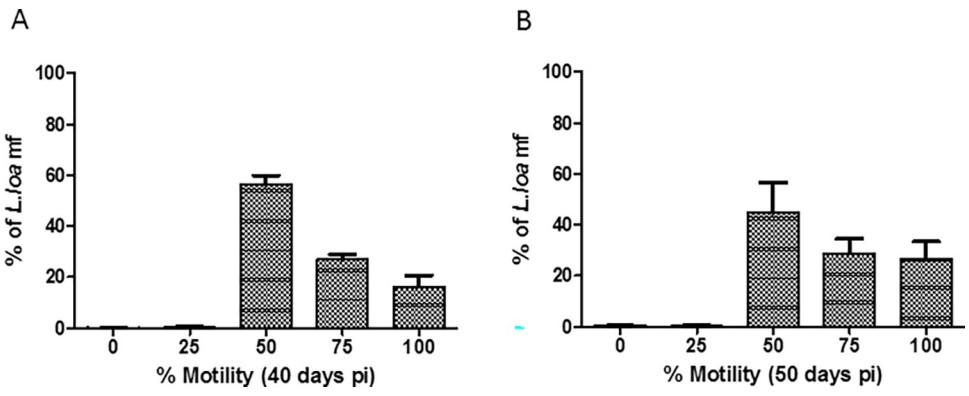

**Fig 6. Percentage motility of peritoneal *L. loa* mf in gerbils.** Day post infection (A) 40 days, and (B) 50 days.

The BALB/c mice are a short-term model for *O. ochengi* mf. Previously, we showed that these mice were not able to retain *O. ochengi* mf for 30 days and some mf were only detected 15 days post-infection [22]. Therefore, in this study, mice were sacrificed 8 days post-infection to improve mf recovery. At this time point, there was a recovery of 0.83% in the earlobes of the control animals, suggesting that recovery could be further improved by analyzing the infected mice earlier. Contrary to the mice, the gerbils had a sustained susceptibility to *O. ochengi* mf with recoveries from the earlobes ranging from 0.75–7.09% between 25–50 days post-infection. The recovery in gerbils is comparable to that of hamsters which had an average recovery of 3.90% after 30 days of infection [22]. Our experience is that the survival of *O. ochengi* mf in mice also varies between the batches of mf isolated, most probably pointing to a cattle-host effect on the mf. Recently, it was shown that cattle-related factors affect the survival of male *O. ochengi* worms in mice and gerbils [32]. Interestingly, IVM cleared all *O. ochengi* mf in both mice and gerbils, while significantly higher parasite numbers were recovered from the untreated animals, thus validating both hosts for microfilariae studies.

The BALB/c mice were equally a short-term model for *L. loa* mf with the best recovery just 2.5 days post-infection (Fig 2A). A significant decrease in mf numbers was noticed as number of days post-infection increased. Similarly, a decrease in mf over time has been reported in the peripheral blood of *L. loa*-infected BALB/c mice [29]. This may be caused by a strong immune clearance of the mf. Use of immunosuppressants, blocking cytokine signaling (IL-4, IL-5 and IFN-γ), general immune system deficiency and removal of the spleen improves susceptibility of host to *L. loa* [27–29,33]. In a comparative study with WT, splenectomized and SCID (lacking mature B and T cells) mice, *L. loa* mf survived best in the SCID mice, and least in the WT, demonstrating a crucial role of the adaptive immune system [29]. Thus, in this *L. loa* mf model with WT mice having an intact immune system, the role of the adaptive immune system could have been assessed. The lung tissues were the preferred homing site of the *L. loa* mf as most of the mf were found there (Table 2). Thus, focus should not only be on the cardiac or peripheral blood for analysis of surviving *L. loa* mf. With respect to number of mf injected, cardiac blood had the least recovered (4.24%) compared to heart and lung tissues with 11.07 and 10.64–14.00%, respectively, 2.5 days post-infection. The *L. loa* mf in the lung and heart tissues are most likely sequestered in the blood vessels. By day 8, the percentage recovery was 0.08, 0.09 and 0.66% in cardiac blood, heart and lung tissues respectively, indicating ~8 times more likelihood to find *L. loa* mf in the lung tissues than in cardiac blood and tissues. The lung tissue is therefore a valuable site for the analysis of the effect of slow acting drugs on *L. loa* mf in BALB/c mice.

The *L. loa* mf BALB/c model is an indispensable tool for the development of new and safe macrofilaricides by counter screening their microfilaricide activities. Novel onchocerciasis drugs must be counter-screened against *L. loa* mf to avoid severe adverse events [2]. Although, the time period from IVM treatment to mice sacrifice is just 48 hours, this duration is still relevant as it was reported that the mean time between IVM intake and onset of severe adverse events in people co-infected with *O. volvulus* and *L. loa* is 24 hours [34].

Intraperitoneal infection of gerbils with *L. loa* mf led to sustained establishment with 33.69% of the mf administered present 50 days post-infection, albeit in the peritoneum. Adult *L. loa* reside in the human subcutaneous tissue producing mf which migrate to the blood. It was expected that *L. loa* mf delivered intraperitoneally will also enter the bloodstream. The *L. loa* mf were rarely detected in the blood of the gerbils. The direct intravenous route was not considered because the tail veins of gerbils are not visible making the procedure cumbersome. At the moment, we are working on intravenous injection of *L. loa* mf using the external jugular vein after surgery [35].

The human dose of IVM is 150 µg/kg body weight through the oral route. Using this dose and route for both mice and gerbils resulted in complete elimination of *O. ochengi* mf in the earlobes (Fig 1), similar to our previous observation in hamsters [22]. However, the same dose led only to decrease in mf motility and a non-significant reduction of number of mf in the peritoneum (Fig 4). However, increasing the dose of IVM from a single dose of 150 µg/kg to two doses of 0.5 mg/kg body weight led to a reduction of *L. loa* mf in the peritoneum by 34.24% and 55.44%, respectively. This suggests that increasing the dose of IVM could lead to a significant decrease in *L. loa* mf contained within the peritoneum. A similar dose of 1 mg/kg IVM given to mice, resulted in a significant reduction in the number of *L. loa* mf in the lungs and in the blood (Fig 3), akin to the results obtained with 10 mg/kg. This indicates that the difference in the effect of IVM may depend on which compartment the mf are found and/or the bioavailability of the drug. IVM (1 mg/kg) resulted in 81.63% reduction in *L. loa* mf in cardiac blood of mice, which is similar to ~92% reduction obtained in a CB.17 SCID mice model [29].

A gerbil mf co-injection model was developed to permit the parallel testing of drugs on both *O. ochengi* and *L. loa* mf. Significant numbers of both mf persisted in the gerbil up to 50 days post-infection, however, the mf motility in the peritoneum decreased. With our ongoing efforts to deliver the *L. loa* mf directly into the blood, it is conceivable that the motility of the injected mf will improve given that this is a more natural niche. We will make efforts to simultaneously infect BALB/c mice with both mf and validate the model with IVM. This will give both short-term BALB/c mice and long-term gerbil co-injection models.

The gerbil model for both *O. ochengi* and *L. loa* mf appears more attractive as there is longer sustenance of the mf in these animals compared to mice. However, while the *O. ochengi* mf are well established in the skin of the gerbils as in humans, the gerbil *L. loa* mf peritoneal model falls short in comparison to the natural blood dwelling mf in humans. Therefore, this model needs to be further developed and validated. The main challenge with the development of this model is having a sustained source of human blood with high loads of *L. loa* mf. It is very important to have donors with loads of ≥15,000 mf/ml to be able to isolate enough mf to inject, especially when administering 40,000 mf intravenously per animal. We have experienced serious decrease in the mf loads of donors which has considerably affected advancing our studies further. In the future we will make sure that we recruit a larger repertoire of donors, so one can easily switch donors if their mf load drops. An alternate source of *L. loa* mf is the baboon model where mf load can be sustained by injecting the infective L3 stage into the animal [33]. Future experiments aimed to better develop this model will focus on intravenous injection of *L. loa* mf isolated from human blood followed by validation with orally administered IVM at 150 µg/kg body weight. If this IVM dose does not result in a significant decrease in *L. loa* mf load, 1 and 10 mg/kg body weight could be considered as was done using the mice model. Furthermore, mf analysis will not only be limited to cardiac blood and lung tissue; the heart tissue will also be considered because it was noticed that 8 out of 9 mice had mf in heart tissue on day 8 whereas only 5 of 9 still had mf in the lungs tissue (Table 2).

Although there has been a strong interest of developing small animal models for the discovery and validation of onchocerciasis macrofilaricidal drugs, small laboratory animal models testing microfilaricidal drugs are still relevant. So far, the gerbil is the only immunocompetent small laboratory animal reported for testing drugs against adult *O. ochengi* male worms [20,21]. To the best of our knowledge, no such immunocompetent model has been reported for the adult female *O. ochengi* worms. Accordingly, the models described here offer a good throughput platform to screen for new microfilaricides and to counter-screen for macrofilaricides. Moreover, these models can be used in onchocerciasis research to better understand the dependence of parasite survival on host immune responses, parasite-host interaction, which will also contribute toward the continued efforts to eliminate filarial diseases in humans.

In summary, when comparing these two models, the BALB/c *L. loa* mf model of 2.5 days is preferable to the long-term gerbil peritoneal model since the mf are in the blood which is the natural infection site in humans. However, such a model is not suitable for slow acting drugs which will require longer exposure time periods to be effective. Thus, for such drugs the gerbil *O. ochengi* mf model is preferred over the BALB/c model because it has a better recovery, and sustained susceptibility for at least 50 days post-infection.

In this study, immunocompetent small laboratory animal models were successfully developed to support the survival of microfilariae for both onchocerciasis and loiasis studies. The BALB/c mouse was developed as a short term model for *O. ochengi* mf (8 days) and *L. loa* mf (2.5 days). The gerbil was established as a long term model for single and co-injection with both mf for at least 50 days.

## Acknowledgments

We thank Gamua Stanley and the staff of Biotechnology Unit, Faculty of Science, University of Buea, Cameroon for their technical support.

## Author Contributions

**Conceptualization:** Judy Sakanari, Sara Lustigman, Fidelis Cho-Ngwa.

**Data curation:** Rene Bilingwe Ayiseh, Glory Enjong Mbah, Emmanuel Menang Ndi.

**Formal analysis:** Rene Bilingwe Ayiseh, Glory Enjong Mbah.

**Funding acquisition:** Judy Sakanari, Sara Lustigman, Fidelis Cho-Ngwa.

**Investigation:** Rene Bilingwe Ayiseh.

**Methodology:** Rene Bilingwe Ayiseh, Glory Enjong Mbah, Elvis Monya, Emmanuel Menang Ndi.

**Writing – original draft:** Rene Bilingwe Ayiseh, Glory Enjong Mbah, Fidelis Cho-Ngwa.

**Writing – review & editing:** Rene Bilingwe Ayiseh, Glory Enjong Mbah, Judy Sakanari, Sara Lustigman, Fidelis Cho-Ngwa.

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
