## [Decision Letter · Decision Letter 0]

5 Aug 2022

Dear Dr Ayiseh,

Thank you very much for submitting your manuscript "Development and validation of small animal models for onchocerciasis and loiasis microfilaricide discovery" for consideration at PLOS Neglected Tropical Diseases. As with all papers reviewed by the journal, your manuscript was reviewed by members of the editorial board and by several independent reviewers. In light of the reviews (below this email), we would like to invite the resubmission of a significantly-revised version that takes into account the reviewers' comments. 

We cannot make any decision about publication until we have seen the revised manuscript and your response to the reviewers' comments. Your revised manuscript is also likely to be sent to reviewers for further evaluation.

Sincerely,

Abdallah Samy

Section Editor

Reviewer's Responses to Questions

**Key Review Criteria Required for Acceptance?**

**Methods**

-Are the objectives of the study clearly articulated with a clear testable hypothesis stated?

-Is the study design appropriate to address the stated objectives?

-Is the population clearly described and appropriate for the hypothesis being tested?

-Is the sample size sufficient to ensure adequate power to address the hypothesis being tested?

-Were correct statistical analysis used to support conclusions?

-Are there concerns about ethical or regulatory requirements being met?

Reviewer #1: The study objectives are clearly stated, and the study design is acceptable given the constraints the Loa loa and Onchocerca ochengi models come with. It is great to see some exploration of O. ochengi and L. loa mf location in infected gerbils (Table 1) and mice (Table 2). Sample size and statistical analysis are appropriate. No ethical or regulatory requirements concerns.

Reviewer #2: The objectives and design of the study are clearly stated and there are no concerns with regard to the sample size, statistical analysis or ethics. However, the material &methods section needs in parts some more information: 

- Please indicate that you used female and male rodents. 

- Indicate the needle gauge size that was used to inject mf. 

- Specify the vehicle used for oral ivermectin treatment (1% DMSO in aqua dest?). 

- Please indicate how you calculated the frequency motility of the microfilariae shown in Fig. 4 (how does a motility of 50% or 75% look like).

One major limitation of the L. loa jird study is that mf were not intravenously injected. Injection of L. loa mf intravenously in jirds using the vena saphena should be doable without anesthesia (in comparison to the jugular vein). This would allow a direct comparison to the i.v. injection of mice and immobile L. loa mf are likely to be cleared from the jird peripheral blood, which better resembles the human host. Thus, you should repeat the L. loa jird experiment with ivermectin treatment after i.v. injection of L. loa mf, which likely results in a better clearance of mf.

For the co-infection experiment (please call it a mf co-injection experiment), human whole blood containing L. loa mf were i.p. injected in gerbils (line 180). The injection of human whole blood likely induces immune responses that affect the response to mf as well. Why did the authors chose this method? Does the injection of purified L. loa mf followed by O. ochengi mf result in a similar outcome?

The different ivermectin treatment regimens and concentrations used do not allow a direct comparison of the models to assess the susceptibility of the microfilariae of both filarial species to ivermectin. This is a major limitation of the study. The authors should at least discuss this limitation of their study and compare theirs doses used with previously published concentrations of ivermectin in rodent models.

**Results**

-Does the analysis presented match the analysis plan?

-Are the results clearly and completely presented?

-Are the figures (Tables, Images) of sufficient quality for clarity?

Reviewer #1: The results are clear and coherent; however I would suggest to add a little more information on the graphs themselves as it gets confusing. For instance, for Figure 1 it would be good to mention in the y axis that in panel A this is in gerbils and in panel B in mice. This would avoid to always refer to the legend to remember which one is which. Same suggestion for Figure 6.

Reviewer #2: The data is clearly presented. Please indicate the gender of the animals used in all figure legends. 

Table 2 indicates that you have indeed on average the highest L. loa mf counts in the lung tissue, but only 5 out of 9 mice had mf in the lung tissue at day 8 after mf injection. In contrast, 8 of 9 mice had still mf present in the heart tissue at that time point. So in order to confirm presence of mf at later time points in mice, the heart tissue may be a better indicator. Do you have data from the mf burden from the heart tissue at 5 and 6 days after mf injection (Fig. 2A, B)? This should be added to the results section. Additional analyses of L. loa mf from the heart tissue of mice may allow the assessment of drugs that lead to a slower L. loa mf removal as ivermectin in mice. This should be discussed.

**Conclusions**

-Are the conclusions supported by the data presented?

-Are the limitations of analysis clearly described?

-Do the authors discuss how these data can be helpful to advance our understanding of the topic under study?

-Is public health relevance addressed?

Reviewer #1: Four main points of concern in this section:

1) After 30 days infection, only 1% O. ochengi mf (232.5 mf recovered on average for 22725 mf inoculated, Table 1) were recovered in ear lobes compared to 5% after 14 days – why has D14 not been chosen over D30 for IVM challenge studies?

2) Due to the rapid elimination of Loa mf in immunocompetent animals (almost no mf left in the lungs after 8 days, Figure 2), a very short therapeutic window is therefore left for testing drugs for their effects on L. loa mf. It would be interesting to investigate whether the number of injected mf can be increased so blood circulating microfilaremia could be the parameter of choice when it comes to evaluate mf killing. Mf localised in the lungs veins could just be trapped there and therefore not reflect true circulating and mobile mf. 

3) As outlined by authors, the co-infection model still needs to be optimized: mode of infection (IV injection to be validated for Loa mf), mf loads (it might be judicious to increase mf loads), IVM dose and injection site (perhaps high dose IVM 5mg/kg subcut as in ref 29), readout time point (might need to be brought forward, especially in the context of co-infections). 

4) Have authors tested other anti-filarial drugs? It would be good to confirm that macrofilaricidal benzimidazole drugs such as flubendazole or oxfendazole don’t have any off-target effect on L. loa mf loads.

Reviewer #2: There is a need for new animal models to test drug candidates for filarial diseases. Especially a better understanding of drug efficacy against microfilariae of L. loa and O. volvulus is important, as indicator for the safety profile in future clinical trials. Thus, this study is of relevance to the field. This aspect is clearly presented in the discussion. However, limitations of the study, such as the different ivermectin doses and regimens have to be mentioned. Furthermore, the authors should highlight why it is important to have immunocompetent animal models to assess drug efficacy. There are several publications that indicate that efficacy of drugs such as benzimidazoles, ivermectin and DEC are supported by the immune system.

**Editorial and Data Presentation Modifications?**

Reviewer #1: Minor revisions needed – see conclusions section for more details regarding requests

Reviewer #2: The first two sentences of the abstract and the author summary are exactly the same. The author summary should be rewritten, so that it does not duplicate the abstract but highlights the importance of this study in a broader context.

Indicate in the introduction (line 109) why the use of immunocompromised hosts is a potential issue for preclinical research.

In general, injection of microfilariae should not be called an infection. Please change this accordingly throughout the manuscript. E.g. line 106 “7 days post-infection” to “7 days post mf injection” to clarify that no adult worm infection was present. Line 167, 172 etc.

Line 374: change to “…validating both HOSTS for microfilariae studies”.

**Summary and General Comments**

Reviewer #1: Developing in vivo drug screening models for filariasis has been (and still is) a challenge. There are several models available right now but many if not all need to rely on modifications brought to the host immune system (immunodeficiencies, splenectomies, specific depletion treatment) or to the use of surrogate parasite species. The authors have therefore concentrated their efforts in using immunocompetent animals in the context of Loa and Onchocerca infections. Despite some limitations (use of O. ochengi, IP injections instead of IV for the gerbil model), they obtained interesting results (especially in the context of IVM treatment) which could lead to some potential for their models to be used for drug screening. However, some aspects would still need to be explored in more details and optimized (mf loads, readout time point, readout parameters, co-infection model). It would be good if the authors could address the raised concerns in the conclusions section of the review before a final decision is made on the manuscript acceptance for publication.

Reviewer #2: The present manuscript describes the potential of immunocompetent BALB/c mice and gerbils that are injected with mf of L. loa or O. ochengi as model to assess the microfilaricidal efficacy of drug candidates. There is a need of such models, as the present small rodent models mainly use immunocompromised mouse strains despite the fact that the immune system may contribute to drug efficacy. One of the major concerns is the intraperitoneal injection of L. loa mf containing whole blood in jirds. The injection of human blood likely induces pro-inflammatory immune response. Furthermore, as seen by the results, intraperitoneally injected mf are not migrating into the peripheral blood and are not cleared by ivermectin treatment. Thus, this experiment should be repeated with i.v. injection of L. loa mf. Another issue is the use of different doses and regimens of ivermectin for the different experiments, which limits the comparisons of the different outcomes. As an additional control, a drug that is mainly macrofilaricidal, such as oxfendazole, should have been once validated.

PLOS authors have the option to publish the peer review history of their article (what does this mean?). If published, this will include your full peer review and any attached files.

Reviewer #1: No

Reviewer #2: No
---

## [Decision Letter · Decision Letter 1]

2 Nov 2022

Dear Dr Ayiseh,

Thank you very much for submitting your manuscript "Development and validation of small animal models for onchocerciasis and loiasis microfilaricide discovery" for consideration at PLOS Neglected Tropical Diseases. As with all papers reviewed by the journal, your manuscript was reviewed by members of the editorial board and by several independent reviewers. The reviewers appreciated the attention to an important topic. Based on the reviews, we are likely to accept this manuscript for publication, providing that you modify the manuscript according to the review recommendations. 

Sincerely,

Abdallah Samy

Section Editor

Reviewer's Responses to Questions

**Key Review Criteria Required for Acceptance?**

**Methods**

-Are the objectives of the study clearly articulated with a clear testable hypothesis stated?

-Is the study design appropriate to address the stated objectives?

-Is the population clearly described and appropriate for the hypothesis being tested?

-Is the sample size sufficient to ensure adequate power to address the hypothesis being tested?

-Were correct statistical analysis used to support conclusions?

-Are there concerns about ethical or regulatory requirements being met?

Reviewer #1: NA

Reviewer #2: (No Response)

**Results**

-Does the analysis presented match the analysis plan?

-Are the results clearly and completely presented?

-Are the figures (Tables, Images) of sufficient quality for clarity?

Reviewer #1: NA

Reviewer #2: (No Response)

**Conclusions**

-Are the conclusions supported by the data presented?

-Are the limitations of analysis clearly described?

-Do the authors discuss how these data can be helpful to advance our understanding of the topic under study?

-Is public health relevance addressed?

Reviewer #1: NA

Reviewer #2: (No Response)

**Editorial and Data Presentation Modifications?**

Reviewer #1: NA

Reviewer #2: (No Response)

**Summary and General Comments**

Reviewer #1: Thanks to the authors for having provided some extra information regarding the points I rose. I however still have a couple of requests before any decision regarding manuscript acceptance can be made:

1) I really appreciated the justification points made for the study limitations in the response to reviewers’ section. I believe some interesting points (the challenges in sourcing more Loa mf, the injection of mf in human blood, the controversy around the analyses of L. loa mf from the heart tissue or the dose of IVM used) would benefit from being acknowledged/discussed in the discussion section of the manuscript. The limitations of the models and the next development steps should be clearly addressed.

2) With all these study limitations and the experiments not having been repeated to confirm the results, I would therefore be prudent on too definitive statements and would temper some of the paper conclusions or statements. For instance, I would remove the “validation” aspect from the paper (title, main text) and make it clearer that some development/validation is still needed (especially in the discussion). The model validation aspect could perhaps be discussed if the interesting PVP/THP work was included in this paper.

Reviewer #2: The authors adequately addressed my previous concerns in their revised manuscript.

PLOS authors have the option to publish the peer review history of their article (what does this mean?). If published, this will include your full peer review and any attached files.

Reviewer #1: No

Reviewer #2: No

Figure Files:

Data Requirements:

Reproducibility:

References

---

## [Decision Letter · Decision Letter 2]

1 Feb 2023

Dear Dr Ayiseh,

We are pleased to inform you that your manuscript 'Development and validation of small animal models for onchocerciasis and loiasis microfilaricide discovery' has been provisionally accepted for publication in PLOS Neglected Tropical Diseases.

Best regards,

Abdallah M. Samy, PhD

Section Editor

Abdallah Samy

Section Editor

Reviewer's Responses to Questions

**Key Review Criteria Required for Acceptance?**

**Methods**

-Are the objectives of the study clearly articulated with a clear testable hypothesis stated?

-Is the study design appropriate to address the stated objectives?

-Is the population clearly described and appropriate for the hypothesis being tested?

-Is the sample size sufficient to ensure adequate power to address the hypothesis being tested?

-Were correct statistical analysis used to support conclusions?

-Are there concerns about ethical or regulatory requirements being met?

Reviewer #1: (No Response)

Reviewer #2: (No Response)

**Results**

-Does the analysis presented match the analysis plan?

-Are the results clearly and completely presented?

-Are the figures (Tables, Images) of sufficient quality for clarity?

Reviewer #1: (No Response)

Reviewer #2: (No Response)

**Conclusions**

-Are the conclusions supported by the data presented?

-Are the limitations of analysis clearly described?

-Do the authors discuss how these data can be helpful to advance our understanding of the topic under study?

-Is public health relevance addressed?

Reviewer #1: (No Response)

Reviewer #2: (No Response)

**Editorial and Data Presentation Modifications?**

Reviewer #1: (No Response)

Reviewer #2: (No Response)

**Summary and General Comments**

Reviewer #1: Thanks to the authors for having considered and addressed all my questions.

Reviewer #2: The authors adressed now all of my previous concerns.

PLOS authors have the option to publish the peer review history of their article (what does this mean?). If published, this will include your full peer review and any attached files.

Reviewer #1: No

Reviewer #2: No

---

## [Editor Report · Acceptance letter]

20 Feb 2023

Dear Prof. Cho-Ngwa,

We are delighted to inform you that your manuscript, "Development and validation of small animal models for onchocerciasis and loiasis microfilaricide discovery," has been formally accepted for publication in PLOS Neglected Tropical Diseases.

Best regards,

Shaden Kamhawi

co-Editor-in-Chief

Paul Brindley

co-Editor-in-Chief
